# Identifying Challenges to the Commercial Viability of Direct Powder Rolled Titanium: A Systematic Review and Market Analysis

**DOI:** 10.3390/ma13092124

**Published:** 2020-05-03

**Authors:** Megan Steytler, Robert Knutsen

**Affiliations:** Centre for Materials Engineering, University of Cape Town, Cape Town 7700, South Africa; megsteytler@gmail.com

**Keywords:** titanium, direct powder rolling, strip thickness, market analysis, applications

## Abstract

A systematic review of factors affecting the viability of direct powder rolling (DPR) as a process route for producing low-cost titanium metal strips was conducted by consolidating performance and process data from published research. Included is a market analysis that was performed by sourcing price points from powder and wrought product suppliers. As a result of the typical oxygen levels (>0.2 wt %) in low-cost powders, the performance of the DPR product is estimated at best to be comparable to ASTM grade 3 and 4 wrought products. Furthermore, evidence supporting chlorine levels >0.02 wt % in low-cost (non-melt) commercially available powders suggest poor weldability, which restricts the application of DPR titanium strips. A comparison of price points for powder and wrought products showed that the potential for commercial viability is likely to exist only for thin gauge strips of <1 mm thickness, as the cost advantage diminishes as the strip thickness increases. Based on the DPR product profile identified in this study (thin gauge, non-weldable, grade 3 or 4), the potential product applications are severely limited. The inability to reliably meet the properties of grade 2 metal strips excludes many uses of titanium metal strips. Consequently, it is emphasized that efforts need to be directed at improving the quality of low-cost powders and developing rolling practices to produce thicker gauge metal strips with desirable properties.

## 1. Introduction

Direct powder rolling (DPR) is the feeding of metal powder into a rolling mill where it is cold compacted into a “green” strip (not yet heat treated). Further processing includes sintering, cold/hot rolling, and annealing to complete densification and optimize mechanical properties. Figure 1 is a schematic example of end-to-end strip processing, from powder rolling to final strip product. The roll compaction of powder is thought to be a more cost-effective, direct route to producing flat product of near final thickness. It is particularly attractive for titanium given that the existing wrought method of melting sponge to ingot followed by rolling to slab, to plate, and finally to thin gauge sheet is an energy and capital-intensive process. Several studies have investigated the operational parameters of DPR, but there has been little assessment of the realization of DPR as a fully operational process producing a commercially viable product. Consequently, the present review is particularly pertinent to inform researchers who are seeking ways to reduce the production costs of titanium metal strips, and the limitations that might arise in choosing the direct powder rolling process route. More specifically, the objectives of the review and the present paper are to address the following. (a) How does the performance of DPR product compare to the performance of product produced via the conventional wrought process route? (b) Does a supply-side market exist to support a commercial enterprise. (c) Lastly, what range of potential product applications could be suitable for DPR products?

## 2. Characterization of Direct Powder Rolled Titanium Metal Strip

### 2.1. Strip Thickness

The thickness of DPR strips is closely limited by the roll diameter. The open literature on DPR does not specifically define the boundaries of the feasible thickness range for titanium green strips. Hence, an attempt has been made to determine this by consolidating research data from individual experiments, which are presented in Figure 2. The data collected were extracted from figures, graphs, and reported results from the earliest records of DPR titanium (1962) up until more recent efforts in 2016. Data for porous strips were excluded, as researchers would not have been aiming for high densities. Furthermore, the data is from vertical, gravity-fed operations only (i.e., no horizontal rolling or screw feeding). A general rule is that the maximum thickness rolled is proportional to the diameter and varies in the range from 0.33% to 1% [1]. Therefore, using a larger roll diameter enables an increase in thickness. Data collected from 17 sources (Figure 2) confirm this, but also show that increasing both the diameter and thickness is at the expense of density. The dashed lines connecting similar symbols in Figure 2 illustrate how rapidly the density of the green strip decreases as the roll gap (and hence strip thickness) is increased for a fixed roll diameter. This is particularly evident for small roll diameters. Assuming that the objective of researchers was to roll high-density strips, then the general negative slope in Figure 2 indicates a thickness-dependent density limit, even when the roll diameters are increased. The limit is approximately indicated by the linear slope drawn on the graph. This behavior is not restricted to titanium and is also evident in the consolidation of DPR density data for other metal powders. Models used to predict the density–thickness relationship for a given roll diameter allow, theoretically, for the attainment of full density by simply allowing for an increase in the roll-separating force (the force exerted on the work roll by the material being compacted between the rolls). In reality, as the rolling machinery is scaled up, the roll forces required to achieve the necessary high density for thicker strips rolled using a larger roll diameter may not be sustainable. However, detailed information in this regard is not available, and this is an area which is encouraged for further research.

The data in Figure 2 are used to estimate the thickness range of the final product. The thickest green strip in this data set is 6.35 mm with 70% green density, rolled on a 1270 mm roll diameter [14] (the largest diameter in the DPR dataset). After several cold rolling and annealing cycles, the final strip thickness was 2.5 mm at 99.7% relative density. This is a 60% thickness reduction. A 50%–70% reduction was found to be commonly cited in the literature for various green strip thicknesses [5,6,10,11,12,13,14,16]. It is assumed that a green strip thickness of 6.35 mm is the upper limit, given that this was rolled with diameters comparable in size to those of large plate mills used in industry [19]. Assuming that 1270 mm rolls are the limit and that at least a 50%–70% reduction is required to achieve near full density, a reasonable expectation for the maximum thickness range of the final DPR product is 2 to 3.2 mm, as shown in blue in Figure 2.

### 2.2. Chlorine Levels

The direct rolling of metal powder is not a new technology. In the late 1950s, Du Pont looked to commercialize a continuous roll compaction process for titanium. They manufactured their own feedstock—powder from a sodium-reduced titanium sponge from the Hunter process—which was particularly suited to DPR due to its morphology and excellent particle interlocking [20]. Initial investigations were deemed commercially promising, and the process was scaled up to tonnage quantities. However, in 1962, Du Pont discontinued powder manufacturing due to unacceptable levels of chlorides, which volatilized during welding and caused a build-up of salt on the welding electrodes. This created an unstable arc that resulted in poor and inconsistent weld quality [21]. The chlorine range in Du Pont’s powder was 0.01–0.05 wt %. In experiments using purer powders that had been manufactured via melting to remove residual chlorides, Du Pont concluded that satisfactory welding could be achieved only at chlorine levels of 0.005 wt % or less, but this target was deemed impractical or too costly at the time [20]. The single greatest barrier to the commercialization of Du Pont’s operation was reported to be the chlorine problem. Du Pont believed that a “marginally weldable product” could not compete with established products on the market [20]. However, Robertson et al. [22] believe that had DuPont persevered, their products were likely to have been adequate for some applications.

Following the discontinuation of Du Pont’s DPR venture in 1962, residual chlorine continued to be cited as problematic. In the 1980s, the prospect of a low-chloride electrolytic sponge garnered attention [23], but efforts were aborted due to engineering or economic issues [24]. Efforts to develop a low cost, electrochemical, or alternative non-melt powder process have continued, but the technology remains developmental. These include the Metalysis FFC Cambridge approach, the MER technique, the CSIRO TiRO method, and the Armstrong process [25].

Today, titanium is extracted predominantly via the Kroll process, and most commercially available powders are derived therein [26]. The Kroll process uses molten Mg to reduce TiCl_4_, and the resultant sponge is known for high chloride impurities. Hence, chlorine problems faced by Du Pont in the 1960s are still relevant today. The purification of a Kroll sponge involves the removal of MgCl_2_ (and other residuals such as Fe) via vacuum distillation. It is the most time-consuming and expensive operation, accounting for about 30% of the final cost of ingot [27]. The refined sponge removed from the reaction vessel varies in quality depending on the degree of purification. It is this intermediate product that is used in the production of powder, whereas to produce wrought product, the sponge undergoes double or triple vacuum arc melting, which removes traces of chlorine completely. The inexpensive titanium powders available today include the following.

#### 2.2.1. Sponge Fines

The sponge is crushed before vacuum arc melting, and those particles that do not meet the size requirement for further conventional processing are termed “sponge fines”. As a by-product, it is a relatively cheap source of elemental titanium [28] but is high in chlorine due to residual salt [27]. Sponge fines have a coarse particle size (typically 180–850 μm, as produced [29] and are limited in application due to the difficulty in grinding ductile particulates to a finer particle size [30].

#### 2.2.2. Hydride–Dehydride Powder (HDH)

HDH offers the advantage of controlled particle size. A starting stock of sponge, scrap, and/or machine turnings is embrittled by heating in a hydrogen atmosphere, resulting in friable TiH_2_, which is subsequently crushed to the desired particle size, and dehydrogenated via heating in a vacuum (TiH_2_ → Ti + H_2_) [28]. HDH powders contain less residual chlorine than the sponge precursor due to the release of Cl from closed pores during crushing [31] and because of the cleaning action of hydrogen removal during the dehydrogenation stage [32], but the risk of oxygen pick-up is greater because of the size reduction process and increased specific surface area [33].

#### 2.2.3. Titanium Hydride (TiH_2_)

TiH_2_ is an intermediate product of the HDH process. It is brittle and therefore not suitable for DPR due to poor particle interlocking, but there is growing interest in compacting TiH_2_ preforms via cold isostatic pressing, followed by dehydrogenation of the compact and thermomechanical work (e.g., forging or rolling to plate).

The chlorine content range for various titanium powders reported in the literature and product quotations (41 sources) is consolidated in Figure 3. Sponge fines have the highest chlorine content (0.08–0.21 wt %). There are two points that have unusually low chlorine, which are labeled as point 1 and 2. They also have unusually small particle sizes (average of 33 μm and <45 μm respectively, compared to the typical, as produced size range of 180–850 μm for sponge fines [29]). The action of crushing to a smaller particle size and consequent release of chlorine, as discussed earlier, is a likely explanation for the low chlorine level of point 1 and 2. However, the downside of such small particle sizes is the increased risk of oxygen contamination, as will be discussed later. TiH_2_ powder has the second highest range of chlorine (0.05–0.12 wt %), followed by the HDH powder range (0.002–0.08 wt % chlorine), which is expected to be lower due to chlorine removal during powder dehydrogenation. The HDH powder could not be separated into that derived from wrought material and sponge, as most sources did not specify the production method. The lowest chlorine content reported for sponge-derived HDH powders is 0.02 wt % [34,35] and 0.023 wt % [36], and chlorine levels below this are assumed to be derived from wrought material, and are therefore not economically feasible. Chlorine l levels as low as 0.02 wt % are also indicated for powders with a PSD (particle size distribution) less than 45 μm. The electrolytic powders investigated in the 1980s show an improvement in chlorine content compared to sponge fines, but levels are above the 0.005 wt % limit identified by Du Pont (dotted blue line), below which no welding problems occur. Furthermore, these powders are no longer commercially available. Of the developmental powders, calcium-reduced hydride powders have relatively low chlorine content (e.g., 0.004 wt %) but were too expensive in the 1980s and did not achieve commercial success [32]. Armstrong powders offer good potential, with chlorine as low as 0.005 wt % or less [37], but these powders are not widely available, and generally have higher oxygen content due to their high specific surface area (a function of their dendritic morphology [38]).

The consolidated chlorine data show that low-cost powders, commercially available today, have chlorine levels that are likely to cause welding problems similar to what Du Pont experienced in their efforts to commercialize the titanium DPR process. HDH, the most commonly used commercial powder, has a chlorine content typically greater than 0.02 wt %. Levels lower than this may be possible for powders with a PSD <45 μm, as chlorine is released during crushing, but an increase in exposed surface area increases the risk of oxygen contamination.

### 2.3. Oxygen Levels

A critical problem for titanium powder metallurgy is titanium’s affinity for oxygen, as it is detrimental to ductility, which is an important performance characteristic for DPR strips given the need to form it into a final product. Given the high specific surface area of powdered material, the formation of oxide layers on individual particles can lead to unacceptable levels of contamination. This significantly affects the mechanical properties of the final product, particularly tensile ductility [39,40]. Oxygen contents for PM (powder metallurgy) product made via DPR, CIP (cold isostatic pressing), and die pressing were consolidated from 25 sources of literature. Up to 90% of the compacts were derived from HDH powders, and the rest were derived from sponge fines. In addition, oxygen contents for wrought product were consolidated from 52 sources of literature. Figure 4 compares the oxygen content of powder-derived product (blue points) to the maximum allowable oxygen content for each of the ASTM grades (dotted line), as well as the typical oxygen content for these grades (yellow points). It is noted that the wider spread in PM data is influenced by experimental testing, while wrought data represent a well-established and finely controlled commercial process.

Only three PM data points are within the grade 1 ASTM limit, and none match the typical oxygen content of grade 1 wrought product. The lowest PM oxygen contents match the middle, or more commonly, the tail-end of the typical oxygen range of grade 2 wrought product; however, the ability to meet the typical properties of grade 3 and 4 is better demonstrated (Figure 4c,d).

The characteristics of the starting powder impact the initial oxygen content as well as the tendency to pick up oxygen during processing [41]. Consolidated data in Figure 5 show the effect of particle size on oxygen content. Point A [42], an outlier, indicates the level of protection that can be achieved via storage under argon, but whether this can be maintained during the processing of powder to a final product, outside of controlled experimental conditions, is yet to be demonstrated.

The effect of particle size on oxygen pick-up after compaction and sintering is demonstrated in the consolidated data in Figure 6, where the increase in oxygen content represents the difference between the initial measurement versus the oxygen level after sintering is complete. Sintered compacts made from powders of smaller mean/median particle size are shown to have higher maximum increases in oxygen. The highest oxygen increases per particle size are circled. The highest increase for a mean/median particle size greater than 50 μm is only 0.07 wt %, whereas for <50 μm, the highest increase is 0.1 wt % or more. Some points show no increase, but this does not indicate that no pick-up occurred. The scenario exists where powders are already excessively oxidized when the oxygen measurement is taken, and hence there is little further uptake during processing, whereas well-protected powders will have a low starting oxygen content, with most of the pick-up occurring during subsequent processing. Points with 0% increase all had a final oxygen content of 0.4 wt % or greater, which supports this argument.

Consolidated data in Figure 7 show how the relative density attainable for a sintered compact decreases as the median particle size increases. The sintering temperature for this dataset is restricted to >1000 °C to minimize variance. Figure 7 shows that a sintered density of >95% is possible for a compact made from powder of approximately 50 μm. For the same particle size, Figure 8 shows that a final oxygen content of 0.25–0.4 wt % (grade 3 and 4 ASTM specification) is possible. The selection of powder particle size must optimize for both sinterability and oxygen contamination. Using powder with a larger particle size to minimize oxygen is constrained by the reduced degree of sinterability of the compact, as well as potential increased chlorine content due to remnant chlorine trapped in unbroken particulates.

## 3. Mechanical Properties of Direct Powder Rolled Titanium Metal Strip

A comparison was made between the performance, in terms of elongation and ultimate tensile strength, of PM and wrought product. The consolidation of data from literature published between 2000 and 2015 is presented in Figure 9. The ASTM elongation specification for a grade 2 sheet and plate (B265) is given as a reference as well as the typical performance of grade 1 to 4 wrought products.

The non-hydride data (excludes TiH_2_-derived product) are separated into two process routes:DPR followed by sintering only, or sintering + cold/hot rolling + annealing (green markers).CIP/die pressing followed by sintering only (white markers).

The hydride data are separated into two processing routes:TiH_2_ CIP/die pressing + sintering only (blue markers).TiH_2_ CIP/die pressing + sintering + hot roll + anneal (purple markers).

The TiH_2_ CIP + sinter + hot roll + anneal route (purple) is an alternative route to producing flat product (in this case plate) with similarities in processing to the DPR route (green). The comparison of these two routes in Figure 9 shows that the best performing DPR sample, with an elongation of 26% (ORNL [43]), outperforms the best performing TiH_2_ CIP sample (ADMA Inc [44]), with 23%. There is a DPR sample with 27% elongation (CSIRO [45]), but this value is only for the rolling direction, and elongation perpendicular to rolling is only 20%. Both the DPR and TiH_2_ CIP route (purple) meet the minimum ASTM specification for grade 2 although, overall, the performance of DPR is better. It is noted that the TiH_2_ CIP dataset is small and from a single source: ADMA Inc. Even so, it is an important dataset due to the ADMA Inc recent commercialization of this route.

The most notable observation in Figure 9 is that typical elongation for grade 2 product (orange) is commonly 5% above the minimum specification and that most of the best performing PM data points (solid circle) fall within the typical grade 3 and 4 performance area. Further observations from Figure 9 are the following: compaction + sintering of hydride powder (blue) outperforms compaction + sintering of non-hydride powder (white); i.e., for a similar processing route, hydride-derived compacts exhibit the highest elongation compared to non-hydride compacts. DPR samples that were only sintered or did not undergo a final anneal exhibit poor ductility (dashed enclosure in Figure 9). As expected, DPR samples generally exhibit higher elongation than non-hydride CIP/press + sinter only samples due to the additional post-sintering cold and/or hot rolling and annealing, which is likely to close remaining porosity. A further assessment of the data in Figure 10 and Figure 11 confirms this and suggests that lower oxygen content in the DPR dataset may also be a contributing factor. Of the data in Figure 9, those points with accompanying oxygen data are presented in Figure 10. The TiH_2_ dataset (blue and purple) is the only set to fall within the limits for both the minimum elongation of 20% and maximum oxygen of 0.25 wt %. By overlaying gradated density data in Figure 11, it is evident that high oxygen and poor density are the reason for the poor performance of the non-hydride CIP/Press + sinter route (white symbols in Figure 10). Many of the DPR samples do not meet the grade 2 oxygen limit, but they do meet the grade 2 elongation requirement, which is possibly due to the high relative densities, as observed in Figure 11. The DPR route, by necessity, involves additional processing, and hence the final product has a relative density of over 99% [4,6,9,10,13,14]. The titanium hydride-derived compacts meet both the grade 2 oxygen and elongation specification, which is possibly due to a combination of low oxygen content (<0.25 wt %) and high final density (greater than 98%).

## 4. Wrought Product Price and Market Analysis

For a DPR flat product to be commercially feasible, it needs to be competitively priced compared to its wrought counterpart. Price data were sourced from online import and export records, product lists from distributors, as well as directly via requests for quotations from distributors. The data are from October 2015 to February 2017. By way of comparison, the current 2020 sponge price is on par with the status in 2017, and the indications are that prices over recent years have been stable. The price dataset was filtered to include only products with dimensions similar to DPR product, these being:A width of 660 mm or less (based on the widest mill, used by ADMA Inc, for DPR to date).A width of 120 mm or more. This excludes very thin products such as ribbon.A length of 500 mm or more. This limits price variability from customized products (sold at a premium) or offcuts (sold at a discount).

Cold rolling, as the final processing, allows for a better surface finish and dimensional control [46]. However, to produce a thin gauge product such as foil, the number of cold rolling and annealing cycles is considerably large [47]. The cost effect of this is seen in Figure 12, where the unit price increases exponentially due to the additional processing and related costs of rolling the product into thin gauge sheets.

Figure 13 exhibits the same data but differentiates the country of origin. It is evident that mill products from China are consistently lower in price across the thickness range in question.

The price dissimilarity between product from China and the US and Europe was found to exist because of the following:There is overcapacity and an oversupply of sponge and mill product globally, but particularly so in China, which has seen a slowdown in demand [48].Although China is the largest producer of sponge, most of it is consumed domestically [49]. Of the mill product and sponge that is exported by China, it is predominantly for industrial applications [50], whereas the US or global sponge and mill markets are driven predominantly by aerospace applications [51,52]. Aerospace grade ingot and slab is typically manufactured via double or triple melting (known as VAR [53]), whereas industrial-grade material is often manufactured via melting in a cold hearth furnace (electron beam or plasma arc), which is lower in cost due to the ability to include a higher percentage of scrap [54]. Industrial mill products do not need to adhere to stringent aerospace specifications, which lowers the price. In addition, the industrial titanium market experiences price pressure from other metals competing for industrial applications [55].

Based on the price data and market review, two distinct markets for wrought products were identified:High-grade aerospace product from the US and Europe.Industrial grade product from China.

These two markets are differentiated in the price data in Figure 14.

## 5. Powder Price and Market Analysis

Powder quotations were sourced from manufacturers and distributors in China, the UK, USA, Russia, and the Ukraine. Most of the data are for HDH powder. A minority is for sponge fines. The price data, in Figure 15, are given as a function of powder size (“<μm” indicates that the particle size distribution (PSD) is below this value). The country of origin is indicated by color. Powder from China is the largest sample and is generally cheaper (<$50/kg) than powder sourced from US, UK, and Russian suppliers. In 2012, Qian et al. [56] reported on China’s growing HDH powder industry, possibly off the back of the expanding sponge industry, with HDH powder priced at approximately $15 to $40/kg at the time, which is similar in range to the current prices for the Chinese product observed in Figure 15.

Chlorine is problematic for most of the powders for which prices were sourced. As exhibited in Figure 16, except for four price points (green), it is not guaranteed that the chlorine content is less than 0.015 wt %, the value below which welding is reported to improve. The powder with the lowest chlorine content (0.003 wt %) is the most expensive, at $180/kg. This is in line with German [57], who reported in 2013 that powders with low interstitial levels are priced in the range of $110 to $220/kg. There are only two low-chlorine data points priced below $50/kg, one of which has an unacceptable maximum oxygen content of 0.4 wt %.

## 6. Feasible Price Range for Titanium DPR Product

The price of wrought strip is compared to the current price of powder in Figure 17. As discussed, for US and EU wrought products, the market is geared toward the aerospace industry, where high-quality aerospace grade sponge, as well as tight tolerances, and stringent production and testing controls [58] are likely to account for the higher prices. Although lower cost powders (those under $50/kg) are priced well below US and EU strip prices (allowing for commercially feasible margins), the quality of this powder is unlikely to yield a product with properties equivalent to the US and EU product expectations. As shown in Figure 17 and in conjunction with Figure 15, almost all the powder priced under $50/kg is from China which, as established, is derived from industrial grade sponge and caters predominantly to the industrial market. Comparing Chinese powder prices to Chinese industrial wrought product prices shows that the potential for commercial viability of DPR strips exists for thin gauge strips of less than 1mm thickness, as this is where the feasible profit margin exists for the industrial grade market.

## 7. Potential DPR Product Applications

The pure titanium grades (1–4) are used predominantly in chemical, marine, power generation, and refining industries where excellent corrosion resistance is a critical requirement. Major applications include heat transfer (condensers, shell and tube heat exchangers, and plate and frame heat exchangers) and welded tubing [59]. Oxygen content determines the strength and cold formability. Grade 1 is the purest form, exhibiting the best room temperature ductility and formability but the lowest strength. It is used where maximum formability is required. Grade 2 titanium is considered the “workhorse” for industrial applications. Its combination of strength and formability make it the most popular choice of the unalloyed grades. It is used where formability and corrosion resistance are important and strength requirements are moderate. Grade 3 is a higher strength grade and is moderately cold formable. Grade 4 has the highest strength and hardness, and the lowest ductility and formability. The influence of the interstitial content decreases with increasing temperature; hence, warm forming of the moderately cold formable grades 3 and 4 is common if the required forming strains are not large [60]. However, the forming of grades 3 and 4 is reportedly performed at room temperature, and all four of the commercially pure grades can be satisfactorily welded and machined [61].

Most of the applications for flat products are fabricated from the purer, more ductile, grades 1 and 2. The formability of these grades is a likely factor, given that forming, bending, and stamping are common methods of fabricating flat products into shapes. Excellent corrosion resistance and ease of formability make grade 1 and 2 the main grades of choice [62], especially in the chemical processing industry, where material selection is driven less by strength and more by the ability to resist corrosion, or to transfer heat, such as in heat exchangers. Grade 2 sees dominant use as it satisfies most performance requirements, and the extent of its usage contributes significantly to efficiencies in production and cost-effectiveness (for example, welding consumables are easily available) [63].

Heat exchangers (HEs) are a major industrial application for pure titanium sheets [59,64]. Although the thermal conductivity is lower than other commonly used HE materials, titanium’s excellent corrosion resistance allows for the use of a thinner gauge product [65]. Grade 1 is the grade commonly used for plate heat exchangers (PHEs) [66] (shown in Figure 18). Thin sheet (e.g., 0.5 to 1mm thickness (HISAKA), or 0.4–0.6 mm (HYDAC International)) is press formed into complex corrugated patterns, which increases the effective heat exchange area. Plates are packed between a fixed and a movable steel frame and tightened together with bolts. The edge of each plate is sealed with a synthetic rubber gasket, so no welding is required (alternatively, they are brazed together). PHE plate patterns are usually cold stamped from any metal that can be cold worked [66,67]. Grade 1 and 2 meet the fabrication requirements of reasonably complex patterns in PHEs [60].

Grade 2 is the most widely used material in heat exchangers [68,69]. Its combination of strength, weldability, formability, and corrosion resistance make it the “explicitly preferred” grade [70]. It is typically used as welded and seamless tubing in shell and tube heat exchangers [66] (Figure 19).

Welded tubing offers commercial potential for DPR because the wall thickness requirements (0.4 mm ≤ WT ≤ 2.77 mm [71]) are within the range producible via DPR, and secondly, the tube is fabricated from a coiled strip. In a highly automated process, a coiled strip is fed into a series of dies, which roll and bend it into the tube shape, followed by welding in an inert environment using a non-consumable tungsten electrode or laser technology [71]. The construction of the shell and tube heat exchanger is more complex than PHEs. The welded pipes are bent into U-tubes and bundled together, and the open ends of the tubes are fixed to a tube sheet (labeled in Figure 19). There are various tube expansion methods to ensure a tight joint between the tubes in the tube bundle and the tube sheet. These methods include direct hydraulic expansion (where fluid pressure expands the tube diameter), explosive expansion, and mechanical rolling where a tapered rotary mandrel expands the tube diameter [72]. Consequently, material requirements for these tubes include a uniform response to the applied bending and expansion methods, and good weldability [60]. The product must pass the flaring test set out in the standard specification for seamless and welded Ti and Ti alloy tubes for condensers and heat exchangers (ASTM B338 – 17). Grades 1, 2, and 3 are included in this specification. Grade 3 has also been used for the tube sheet [73].

There is little evidence of the use of grade 4 in heat exchangers. The few existing applications are lightweight plate-fin HEs in aircraft cabin control systems that cool bleed air [74]. Another widely used titanium product is welded piping, which is used in power generation; chemical processing industries; and oil, gas, and petroleum processing [28]. Titanium is less prone to crevice corrosion in ambient temperature seawater, and consequently, it is used extensively in piping in marine and offshore oil and gas operations [75]. The standard specification for Ti and Ti alloy welded pipe (B862-14) is applicable to grades 1–3 and 5, and wall thicknesses of 1.24 mm up to 30mm depending on the outside diameter of the pipe.

Titanium plate or shell and tube heat exchangers, as well as titanium piping, would be particularly suitable for production via DPR for the following three reasons:Unlike aerospace, medical, and pressure vessel applications, the quality standards and expectations for products used in industrial heat exchangers and piping are less stringent, offering a natural market entry point.The dimensional requirements are within the process capabilities of DPR (width <610 mm and thickness <3 mm).Heat exchangers and piping are the most common applications for pure titanium and flat mill product (welded CP titanium tubing is deemed an important intermediate product^60^). Therefore, the market is large enough to warrant commercial potential.

However, the use of DPR product in heat exchangers and piping is problematic due to requirements for good formability and/or weldability. Plate heat exchangers sealed using gaskets do not require welding, but excellent formability is key, warranting the use of high-purity grade 1. For tube heat exchangers, which are commonly made from grade 2, the sheet stock must be weldable and must have sufficient biaxial formability to meet flaring requirements.

## 8. DPR Strip Fabricability

### 8.1. Formability

ORNL [14] investigated the applicability of DPR sheets for use in heat exchangers (HEs). They tested the elongation, bend radius, and dome height, and stamped herringbone and dimple patterns. The elongation performance of the DPR sheet tested by ORNL is shown in Figure 20 (in blue) in relation to other fully processed DPR samples in the consolidated dataset (green), as well as the typical properties of grade 1–4 wrought products. Pairs of longitudinal and transverse elongation are circled.

ORNL samples with both longitudinal and transverse elongation within the grade 2 limit are some of the best performing in the DPR dataset and are technically grade 2 as they meet the minimum elongation requirement. However, the sheet performed poorly in the dome height test, and the low biaxial formability was attributed to the presence of voids. Consequently, the sheet could not be stamped successfully at room temperature (improved formability was found at 300°C and excellent formability was found at 600 °C). Duz et al. [76] concluded that the sub-standard formability of the DPR product made it unsuitable for die-pressed plate heat exchangers. A recommendation for flat-plate heat exchangers was made with reference to Campbell Applied Physics, who required very thin flat sheets (0.2 mm) for a proprietary heat exchanger.

Kapranos et al. [77] claim that because ductility is critical for the fabrication of plate heat exchangers (and for the expansion of tube sheets in shell and tube HEs), the upper range of performance of grade 2 product is preferred. In Figure 20, it is evident that the DPR product is not of typical grade 2 quality (25%–30% elongation) and is certainly not upper grade 2 quality (the lower oxygen content range). Lunde et al. [78] claim that oxygen content is even more important than ductility, and that no more than 0.16 wt % should be present in grade 2 material to avoid microcracks during cold forming. There are only two PM points in the dataset in Figure 4b that achieve 0.16 wt % oxygen, whereas most of the typical grade 2 wrought product samples are well below this, as demonstrated in the distribution in Figure 21.

### 8.2. Weldability

Welding problems due to remnant chloride impurities from the Kroll reduction process were discussed earlier, and it was established that the low-cost, non-melt, Kroll sponge-derived powders available today are still above the chlorine content identified by Du Pont (0.005 wt %), above which welding problems occur. Further problems, not related to chlorine impurities, have also been observed in the welding of powder metallurgy components. Muth et al. [38] recently investigated the ability to fusion weld various low-chlorine containing Ti and Ti-6Al-4V plates made from the following powders:Armstrong powder (low chlorine, developmental powder).HDH powder made from revert scrap (ingot derived and therefore low in chlorine).TiH_2_ (Ti-6Al-4V plate purchased from ADMA with low Cl due to “cleaning” effect of TiH_2_).

Although all plates had a chlorine content less than 0.001 wt % and were consolidated to full density, the weld quality was unacceptable. Weld porosity was attributed to hydrogen gas from water adsorbed on the surface of powder particles, as well as gas-forming sodium or magnesium species, remnant from the reduction of sponge (which is usually removed during triple VAR melting of sponge). Muth et al. ultimately concluded that the PM product could not be reliably fusion welded. Greenfield et al. [79] reported similar porosity problems in the welding of fully-dense Ti-6Al-4V HIPed (hot isostatic pressed) compacts made from low chlorine pre-alloyed HDH powder (i.e., ingot derived). Although the welding of compacts made from spherical pre-alloyed Ti-6Al-4V powder was not problematic, linear porosity (an array of porosity) in the weld zone of the HDH compact was attributed to adsorbed gases, other than hydrogen, on the higher specific surface area of the HDH powder (a six-fold difference in surface area was determined). The as-welded tensile properties of the fully dense HIPed HDH compacts were not “appreciably affected” by the linear porosity, but Greenfield et al. concluded that HDH-derived compacts would not be acceptable for fatigue-related applications. Compacts that were not fully dense (99.9%, which is still very high) were found to be non-weldable due to outgassing in the fusion zone, which destabilized the arc.

The low–chloride containing specimens used by Greenfield et al. [80] and Muth et al. [38] were made from powders that are not suitable for a commercial DPR operation, as they are either too expensive (HDH derived from wrought product), developmental and not readily available (Armstrong), or not compatible with the process (brittle TiH_2_). Low-cost commercially available powders that are suitable for DPR (e.g., sponge-derived HDH) contain higher chlorine levels (as established earlier), and hence welding problems are likely to be more severe that those observed by Greenfield et al. and Muth et al for low-chlorine compacts.

## 9. Conclusions

The profile of DPR product has been narrowed down to the following: non-weldable strip, of up to 2–3.2 mm thickness, matching the typical tensile performance and oxygen content of grade 3 and 4 wrought product. The maximum thickness of 2–3.2 mm is what has been identified as technically feasible, but a thickness of 1 mm or less is likely to be economically feasible based on the powder price and market analysis, as this is where the cost advantage occurs. However, applications for thin gauge grade 3 and 4 product are limited, as these grades are typically selected for thicker product where greater strength is a requirement. The inability of DPR products to meet the upper range of typical grade 2 elongation (>25%) and the lower range of oxygen (0.16 wt %) excludes the largest proportion of potential product applications for titanium. The most critical commercialization barrier for DPR is the limited range of potential applications suitable for the DPR product profile identified in this research. For this reason, it is concluded that the indications at present are that DPR is not a broadly commercially viable process.

Improvements in the feasibility of DPR can be gained through progress in powder production methods and direct powder rolling technologies. It is clear that it is possible to produce appropriate quality powder but significant reductions in cost are required to be able to meet at least grade 2 standards, and thus robust, commercially applicable processes are required to compete with the well-established Kroll process. The limit in strip thickness could be addressed by including roll bonding processes, either in the green state or in the partially sintered state, or perhaps a combination of both. In this way, the DPR strip product can compete more viably with grade 3 and 4 applications.

## Figures and Tables

**Figure 1 materials-13-02124-f001:**
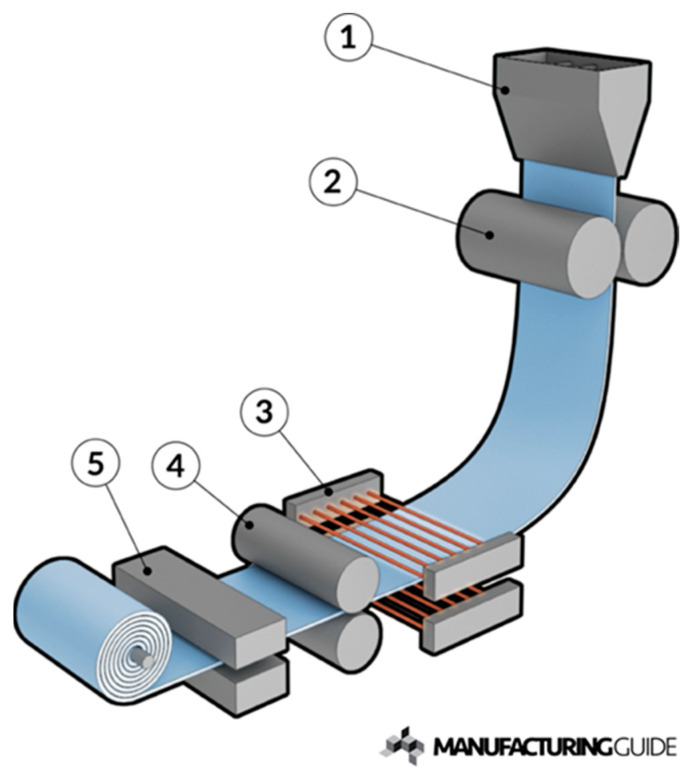
Generic process layout for direct powder rolling. Powder feeder (1), roller (2), sintering furnace (3), additional rolling (4), and cooling (5) (source: https://www.manufacturingguide.com/en/metal-powder-rolling).

**Figure 2 materials-13-02124-f002:**
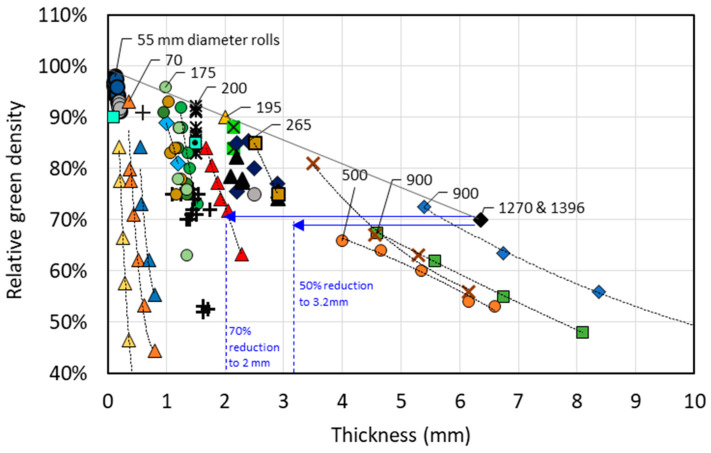
Green density and thickness for titanium and Ti-6Al-4V strips produced using various roll diameters [2,3,4,5,6,7,8,9,10,11,12,13,14,15,16,17,18]. Color and symbol combinations indicate different studies. Numbers next to symbols indicate roll diameters, which only some papers specified. Line drawn across the highest density points shows decreasing maximum density for increasing strip thickness.

**Figure 3 materials-13-02124-f003:**
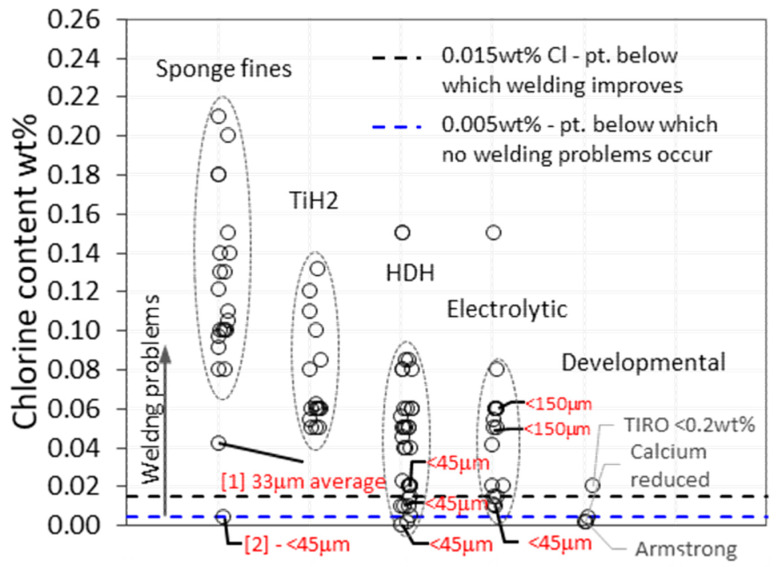
Chlorine content of various titanium powders. Data from literature and direct powder quotations (41 sources). Powder particle sizes labeled in red.

**Figure 4 materials-13-02124-f004:**
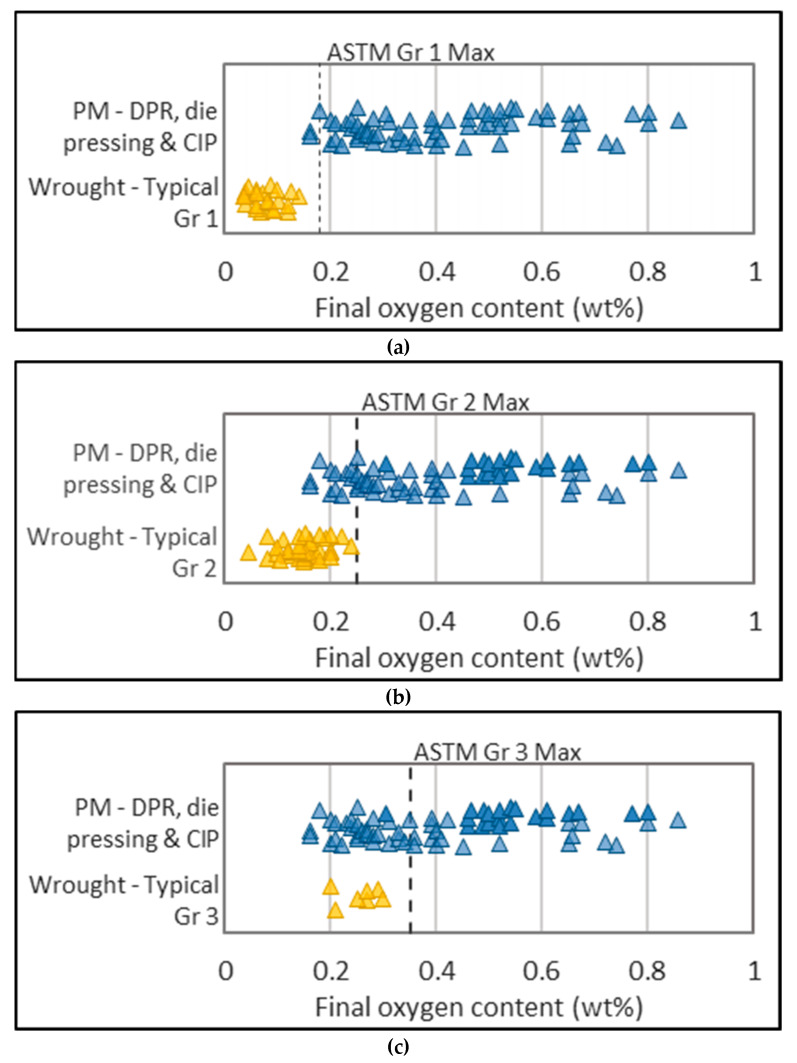
Final oxygen content for titanium powder metallurgy (PM) compacts versus typical oxygen content for wrought product specified to ASTM grades 1–4 (**a–d**).

**Figure 5 materials-13-02124-f005:**
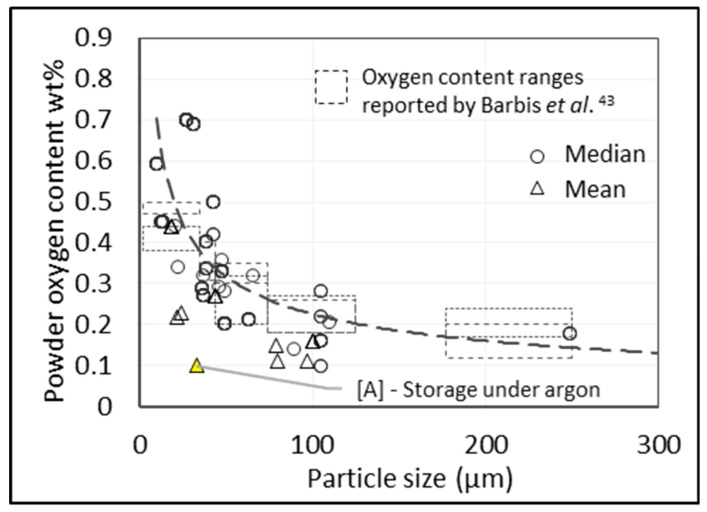
Initial oxygen content of Ti and Ti-6Al-4V powders (sponge fines and hydride–dehydride powder, or HDH) versus particle size. Data points from 13 sources.

**Figure 6 materials-13-02124-f006:**
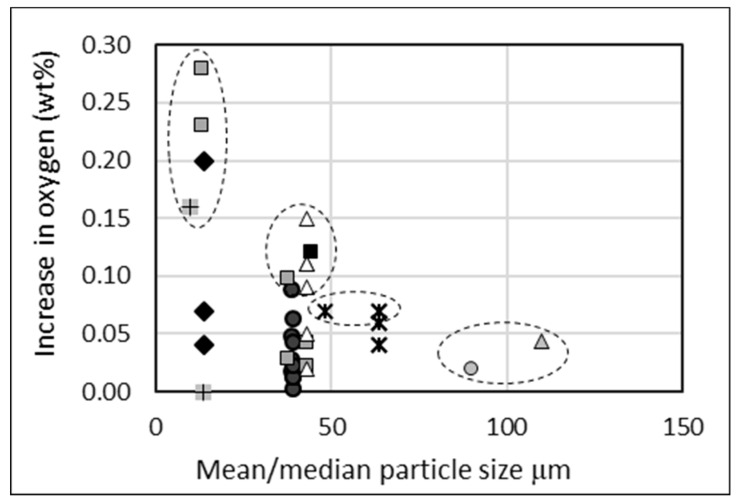
Increase in oxygen content from powder to sintered compact versus mean/median particle size. Sponge fines and HDH powder only, direct powder rolling (DPR), cold isostatic pressing (CIP), and die pressing. 10 sources identified by different markers. Largest increases in oxygen per study are circled to show the maximum increases per mean/median particle size, and regardless of processing/powders used, the increasing risk of oxygen contamination as powder size decreases.

**Figure 7 materials-13-02124-f007:**
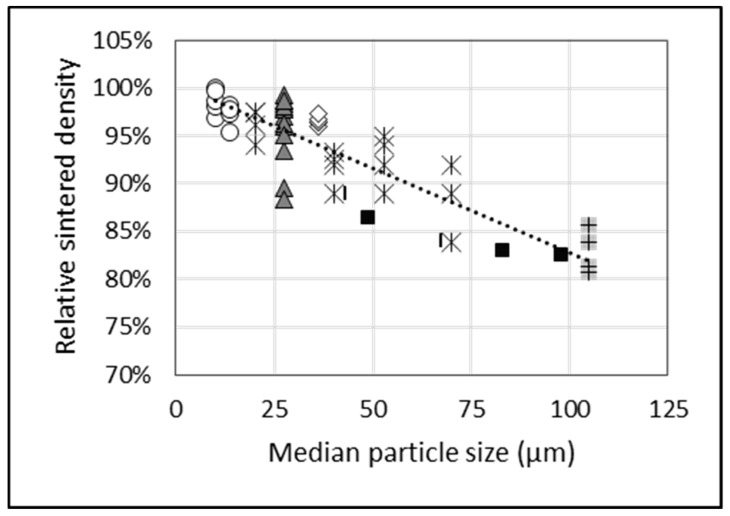
Relative sintered density versus median particle size for cold compacted titanium samples from HDH powder. Sintering temperature ≥ 1000 °C. Compaction pressure ≥ 200 MPa. Data from 6 sources, as indicated by the different symbols.

**Figure 8 materials-13-02124-f008:**
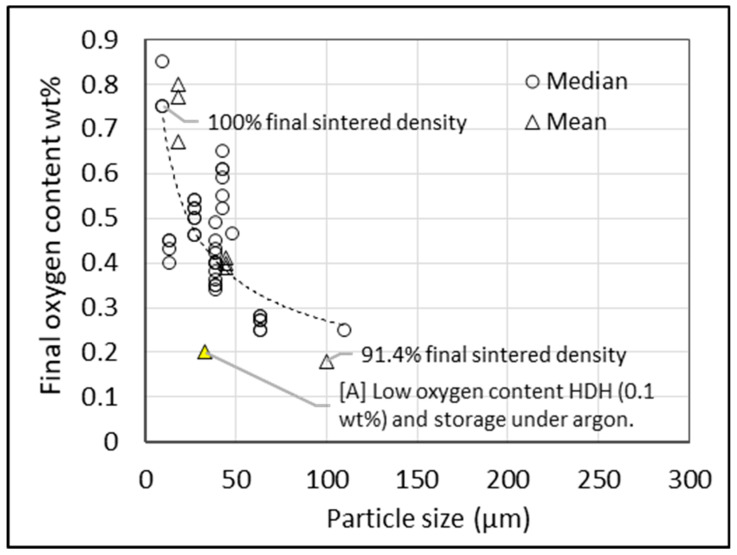
Final oxygen content after sintering of Ti and Ti-6Al-4V compacts versus particle size (compacts from sponge fines and HDH powder). Data from 11 sources.

**Figure 9 materials-13-02124-f009:**
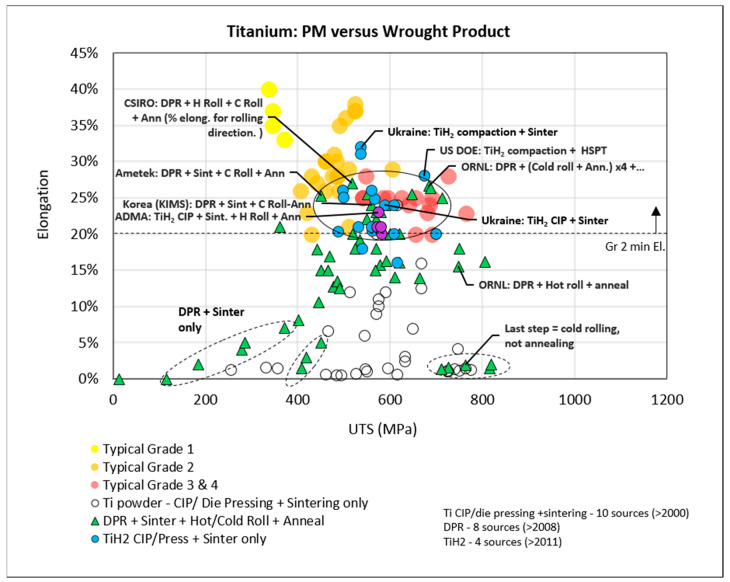
Elongation and ultimate tensile strength of wrought product versus PM compacts from various process routes and feedstock.

**Figure 10 materials-13-02124-f010:**
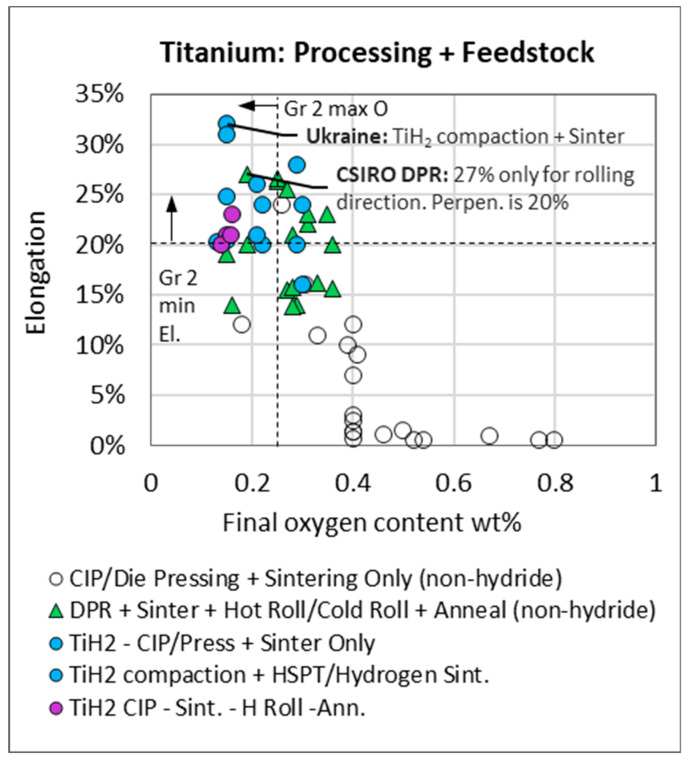
Elongation versus oxygen content for titanium compacts from hydride and non-hydride powder (12 sources).

**Figure 11 materials-13-02124-f011:**
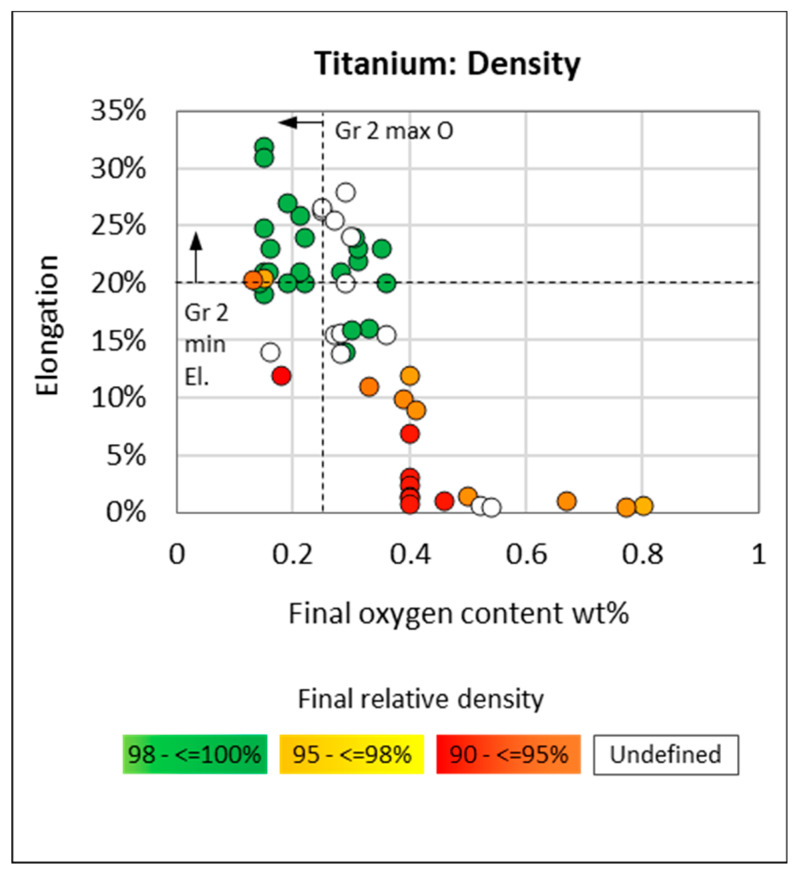
Elongation versus oxygen content for titanium compacts from hydride and non-hydride powder with final relative densities (12 sources).

**Figure 12 materials-13-02124-f012:**
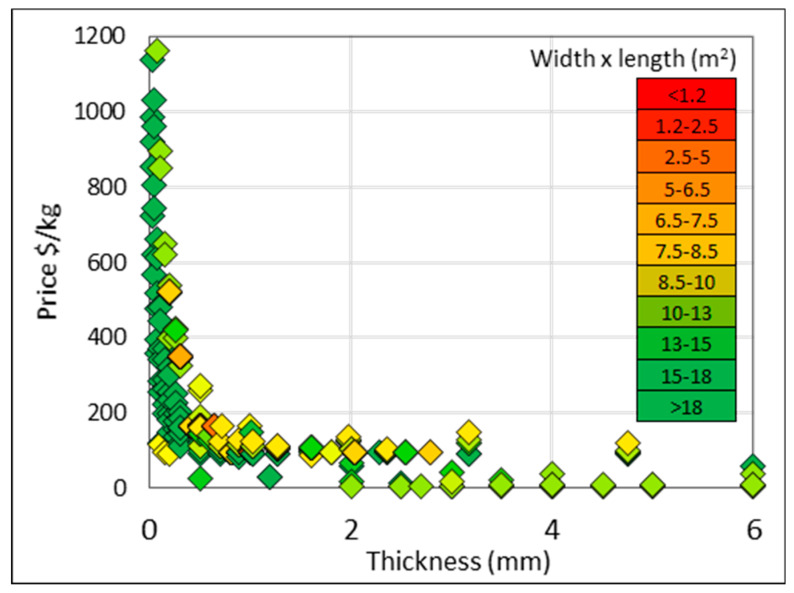
Price data for grades 1–4 titanium strip with width >120 mm and <660 mm, and length >500 mm. Width and length shown as color gradations.

**Figure 13 materials-13-02124-f013:**
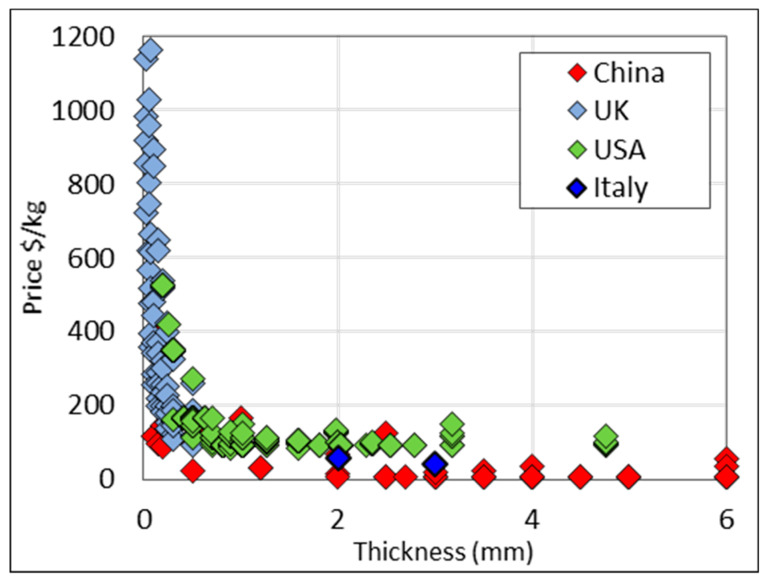
Price data for grades 1–4 titanium strip. Country of origin distinguished by color.

**Figure 14 materials-13-02124-f014:**
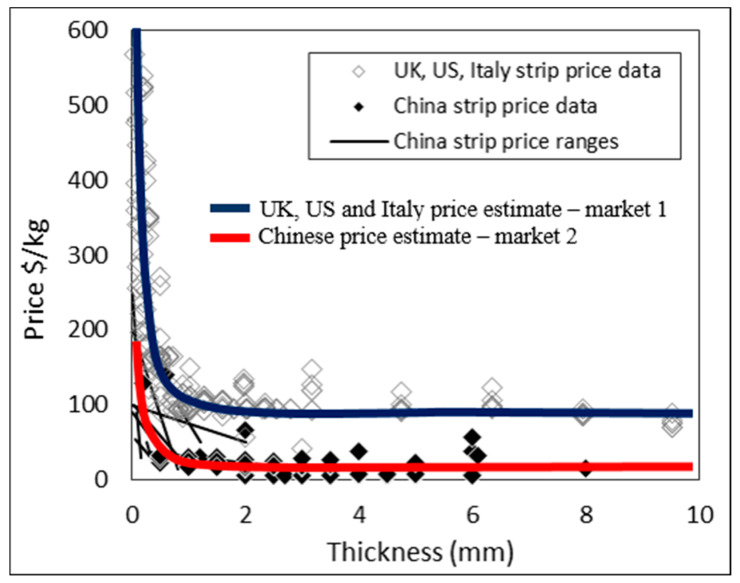
Middle price estimates for grades 1–4 titanium strip for the UK, US, and Italy versus China. Data is only for strip widths >120 mm and <660 mm, and lengths of 500 mm or more.

**Figure 15 materials-13-02124-f015:**
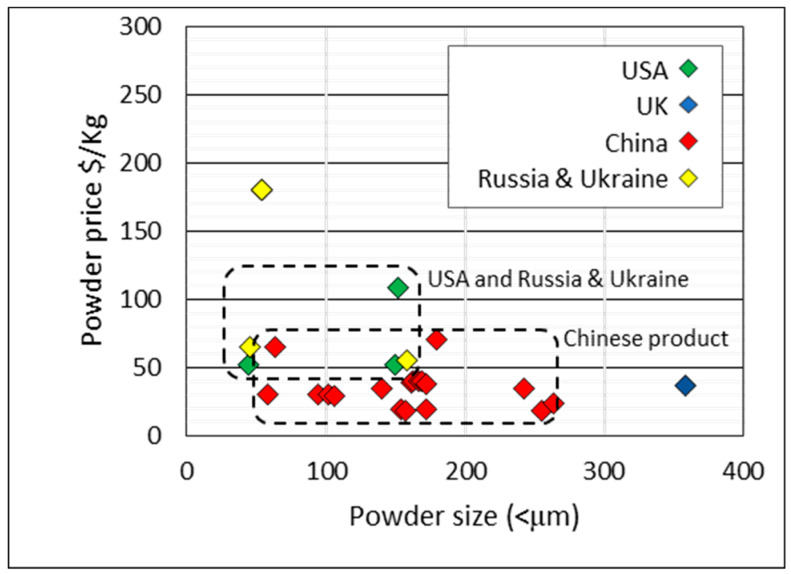
Price per kg of titanium powder (HDH and sponge fines) and powder size (<µm). Country of origin indicated by color.

**Figure 16 materials-13-02124-f016:**
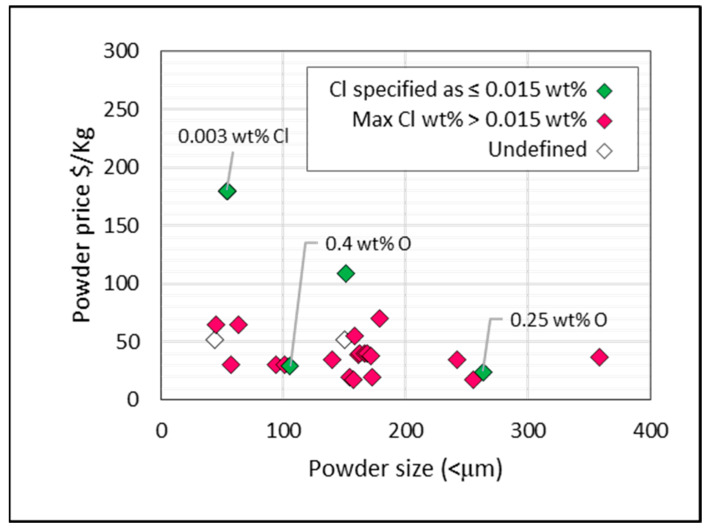
Price per kg of titanium powder (HDH and sponge fines) versus powder size (<µm). Chlorine content indicated by color.

**Figure 17 materials-13-02124-f017:**
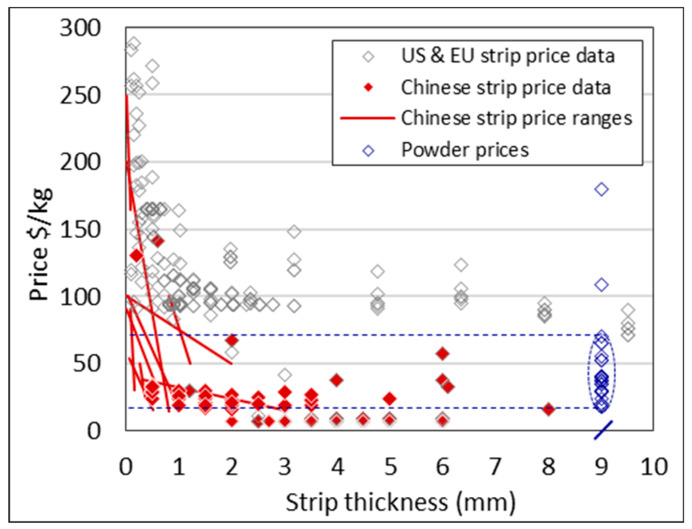
Price of wrought strips from China and the US and EU versus the price of titanium powder (powder prices indicated at/mark on x-axis). The straight red lines indicate a range.

**Figure 18 materials-13-02124-f018:**
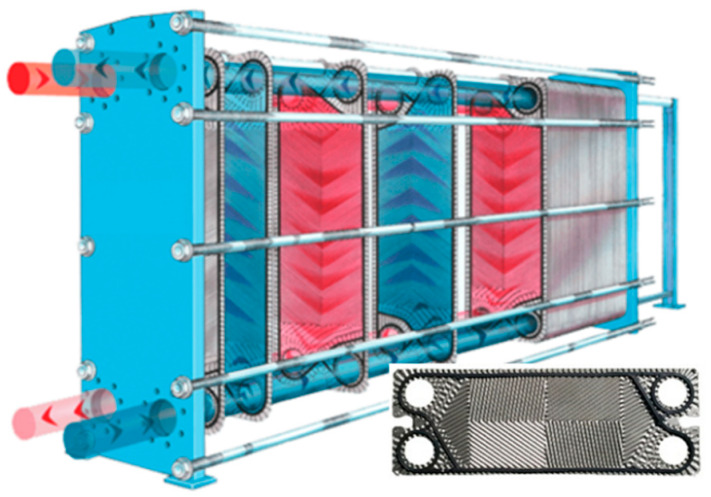
Plate heat exchanger and example of plate pattern (source: Grano Heat Energy Technology Co. Ltd. https://www.grano-heat.com).

**Figure 19 materials-13-02124-f019:**
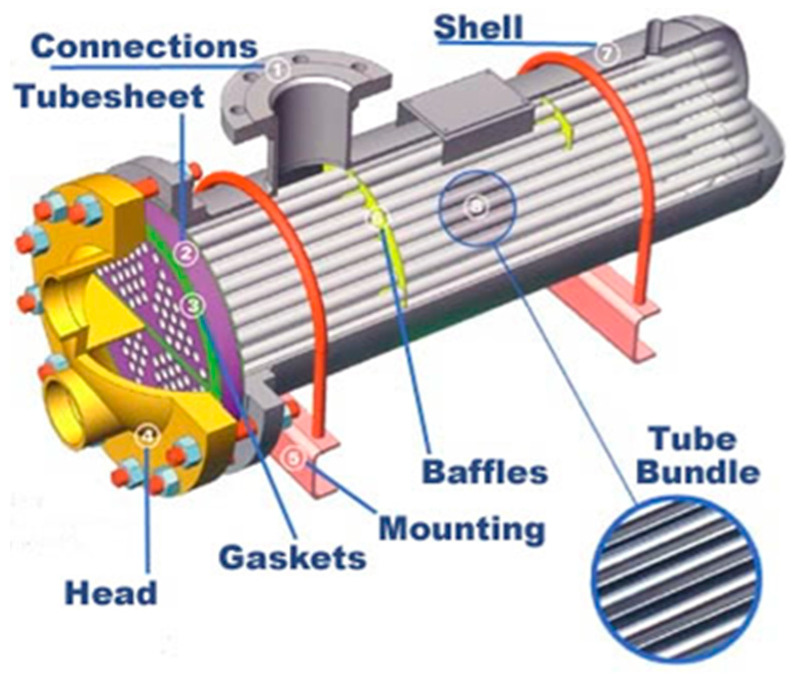
Shell and tube heat exchanger (source: South West Thermal Technologies Inc. http://www.SouthwestThermal.com).

**Figure 20 materials-13-02124-f020:**
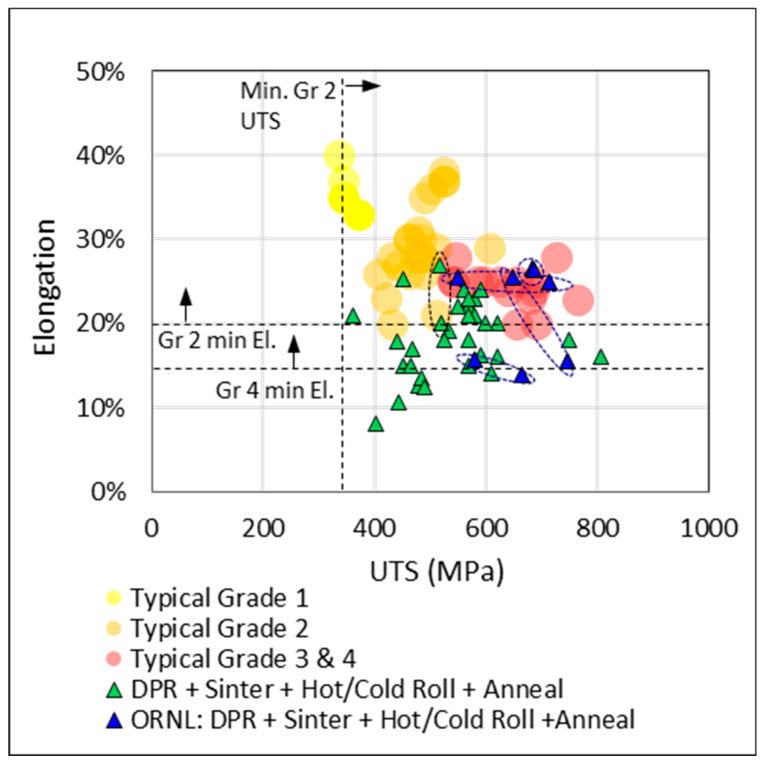
Elongation and UTS of fully processed DPR strip versus typical properties of grade 1–4 titanium. DPR: 8 sources (>2008).

**Figure 21 materials-13-02124-f021:**
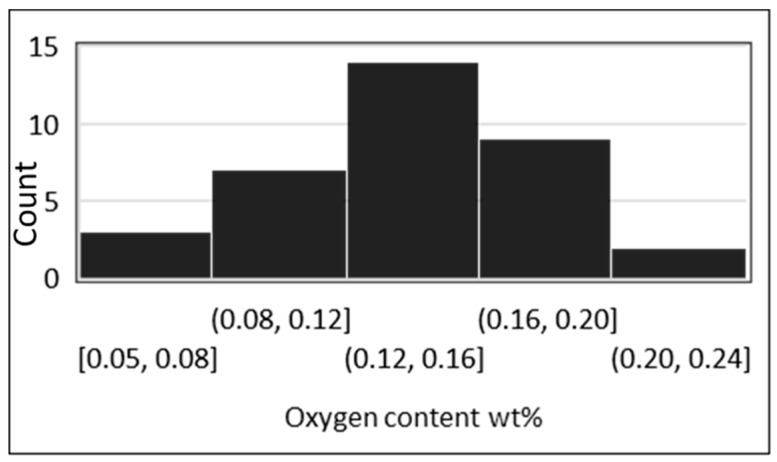
Distribution of typical oxygen content for grade 2 wrought product (ASTM maximum is 0.25 wt %). Sample size = 35 sources.

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
