# Peer review of "Identifying Challenges to the Commercial Viability of Direct Powder Rolled Titanium: A Systematic Review and Market Analysis"

_materials, 2020, doi:10.3390/ma13092124_

Round 1

Reviewer 1 Report

The article contains a lot of interesting information about rolled titanium powders. In my opinion chapter 8. DPR strip fabricability is very general. I would recommend that you consider the following comments.

  1. I would recommend showing a diagram of the Direct powder rolling (DPR) process to better understand the described process.

  2. The possibilities of metal forming (8.1. Formability )of these materials are very interesting and should be supplemented.

  3. The article does not provide the requested information on the machining possibilities of these materials. It is worth adding some information on this topic.

  4. In fig. 21. The “x” axis should be described. Because it is not clear what the values from 0 to 15 represent.

Author Response

  1. We have added a new Figure 1 to illustrate the direct powder rolling process. Reference to Figure 1 in text in lines 31-32 and new figure at 45-46.
  2. No change: the possibility of the DPR product having application in products where formability is required is presented already in section 8.1. Because the production of DPR Ti strip is still in development, data on actual formability of the DPR Ti-strip is not yet available.
  3. No change: machinability of DPR Ti-strip is not relevant.  Because it is argued in our paper that the DPR Ti-strip is limited to very thin strip only, machining is not a viable manufacturing process.
  4. A label has been added to the y-axis of Figure 21 (the reviewer mentions x-axis but in fact they implied the y-axis).

Reviewer 2 Report

The paper represents a significant contribution to our understanding of DPR Titanium. They also highlighted an analysis of the market with the prospect of its real application. It is a significant topic in the current Titanium industry. Overall, this paper is well-organized and will be a good publication. I think, nonetheless, that the manuscript could be improved if the authors could address the comments and recommendations I listed below.

  1. Line 71: Figure 1 could have some modifications. You have numerous symbols represent different roll diameters. Actually, it is not very easy to read. I suggest you can add a legend next to your plot.
  2. Line 118: 2.2.1 is a subtitle from 2.2 aiming to discuss chlorine levels. To have a good connection to the title. I suggest you add some discussion about the connection between sponge fines and chlorine levels.
  3. Line 176: Abbreviation “PM” means powder metallurgy. You should put the full name here since no explanation of this abbreviation before. It will help the audience not familiar with the alloy industry.
  4. Line 222: In Figure 5, what is the meaning of the circles? Legend is needed. The description of symbols is needed.
  5. Line 234: Same problems. Legend is needed. The description of symbols is needed.
  6. Line 267: Could you replace Figure 8 with a higher photo resolution? It is kind of hard to read in the current version.
  7. Line 247-254: Some citations may need here.
  8. Line 302: What’s your price data source or website? Citations may need here. Also, in figure 11 & 12, you may need to mention it.
  9. Line 391: In Figure 16, the fitting of Chinese strip price ranges lines should be reconsidered. I recommend using, at least, parabolic rather than a straight line.
  10. Line 456: Titanium has good corrosion resistance, but it is not immune and can be susceptible to pitting and crevice attack at elevated temperatures. You may consider rewording this sentence.

Author Response

  1. Changed: Comments about Fig 1 (which is now Figure 2 (see lines 76-77).  The caption (lines 78-81) has been modified to improve the explanation of the graph.  Coloured symbols are added to help distinguish between different studies. It is indicated that not all papers specified roll diameters, hence we have not added a legend but left data labels for roll diameters where applicable. We have indicated the reason for the line drawn through the max densities per strip thickness.
  2. Changed: We have added a comment about chlorine occurrence in the sponge fines (line 128).
  3. Changed: added (powder metallurgy) to explain PM - see line 183.
  4. Changed: Clarification sentence added in the figure caption (now figure 6) - see lines 231-233.
  5. Changed: Added comment in caption to clarify (now figure 7) - see line 246.
  6. Changed (now Figure 9): The font for the data labels has been changed to bold and black.  Resolution of the Figure 9 has been improved has been improved (lines 274-275).
  7. Not changed: Lines 256-263 (was 247-254) is simply explaining how we have grouped the data for presentation in the graphs.  It does not relate to any particular citations.
  8. Not changed: Just before line 302 (which is now line 308) we state "Price data was sourced from online import and export records, product lists from distributors, as well as directly via requests for quotations from distributors". Consequently price was sourced from a large and mixed data set and synthesised to present in a meaningful way in Figs 12 and 13 (previously Figs 11 and 12).  We have added a sentence in lines 310-311 to illustrate that the prices in 2020 remain similar.
  9. Changed: The straiight lines are not an attempt to fit the data, but rather they indicate a range.  A note has been added to the caption to indicate as such (see line 402).
  10. Changed: We have modified the sentence (now line 466) to indicate that titanium is less prone to corrosion (and hence not immune).

Reviewer 3 Report

Dear Authors,

The manuscript is a review of the powder rolling of Ti alloys and their comparison to other production processes, discussion of the economic issue and possible applications. The following points were discussed: the thickness of the Ti strip as the density of the material, chlorine removal process, oxygen content influence and its relation to several Ti alloys, the effect of Ti particle size on oxygen content, the influence of the oxygen content on the material’s mechanical properties, the price effect on the strip thickness. Additionally, different industrial applications of Ti were discussed.

The manuscript is well written with logical structure and fine English. However, several minor remarks should be considered:

  1. I recommend to add chemical reactions to the hydrogen removal process described in 2.2.2
  2. Figure 8, please increase the contrast of the image.
  3. Please provide more recent prices for the Ti alloys, i.e. using metal exchange.
  4. I recommend to add one paragraph about biomedical applications of Ti alloys, such as various medical devices, implants, etc.

Author Response

  1. Changed: Reaction has been added (see line 135).
  2. Changed: Now Figure 9 - same comments as for Reviewer 2 (font is bold and black, and resolution is improved - see lines 274-275).
  3. A check on current powder prices shows little change and no significant outliers when compared to the price data consolidated in this review. Lowest prices seen today are still around the $20/kg mark. The purpose of the pricing analysis was to identify the price difference between powder and final wrought product. Since titanium wrought processing is a very well established industry, price fluctuations are very market/demand oriented. There has been little change in Ti prices from 2015 to 2020 (https://fred.stlouisfed.org/series/WPU102505 and https://www.mining.com/wp-content/uploads/2020/03/Titanium-Sponge-Price-1.png). As titanium powder is pegged to the wrought industry due to a common base material (Ti sponge), truly large material differences between wrought product and powder would be as a result of major advancements in powder processing, of which no new publications point to at present. We have added a sentence in line 311 to note stable sponge price.
  4. Not changed: Biomedical applications of Ti alloys were not included in the potential products section because our review narrowed the likely properties and performance of DPR strip to lower-grade industrial applications (these are the arguments that we present). The more stringent biomedical specifications were deemed too specialized, narrowing the commercial scope, and hence were not investigated as a feasible application for a low-cost, non-melt titanium product.  Of lesser importance, but also relevant, is that thin strip metal has less application in biomedical products (the latter are more 3-dimensional and either machined from wrought stock or produced directly from powder using additive manufacturing).